# Unexpected Role of MPO-Oxidized LDLs in Atherosclerosis: In between Inflammation and Its Resolution

**DOI:** 10.3390/antiox11050874

**Published:** 2022-04-28

**Authors:** Cecilia Tangeten, Karim Zouaoui Boudjeltia, Cedric Delporte, Pierre Van Antwerpen, Keziah Korpak

**Affiliations:** 1RD3-Pharmacognosy, Bioanalysis and Drug Discovery, Faculty of Pharmacy, Université Libre de Bruxelles, 1050 Brussels, Belgium; cedric.delporte@ulb.be (C.D.); pierre.van.antwerpen@ulb.be (P.V.A.); 2Laboratory of Experimental Medicine, ULB 222 Unit, CHU-Charleroi, A. Vésale Hospital, Université Libre de Bruxelles, 6110 Montigny-le-Tilleul, Belgium; karim.zouaoui.boudjeltia@ulb.be (K.Z.B.); keziah.korpak@chu-charleroi.be (K.K.); 3Department of Geriatric Medicine, CHU-Charleroi, Université Libre de Bruxelles, 6042 Charleroi, Belgium

**Keywords:** myeloperoxidase, atherosclerosis, inflammation, myeloperoxidase oxidized low-density lipoprotein, specialized pro-resolving mediators, resolvin D1

## Abstract

Inflammation and its resolution are the result of the balance between pro-inflammatory and pro-resolving factors, such as specialized pro-resolving mediators (SPMs). This balance is crucial for plaque evolution in atherosclerosis, a chronic inflammatory disease. Myeloperoxidase (MPO) has been related to oxidative stress and atherosclerosis, and MPO-oxidized low-density lipoproteins (Mox-LDLs) have specific characteristics and effects. They participate in foam cell formation and cause specific reactions when interacting with macrophages and endothelial cells. They also increase the production of intracellular reactive oxygen species (ROS) in macrophages and the resulting antioxidant response. Mox-LDLs also drive macrophage polarization. Mox-LDLs are known to be pro-inflammatory particles. However, in the presence of Mox-LDLs, endothelial cells produce resolvin D1 (RvD1), a SPM. SPMs are involved in the resolution of inflammation by stimulating efferocytosis and by reducing the adhesion and recruitment of neutrophils and monocytes. RvD1 also induces the synthesis of other SPMs. In vitro, Mox-LDLs have a dual effect by promoting RvD1 release and inducing a more anti-inflammatory phenotype macrophage, thereby having a mixed effect on inflammation. In this review, we discuss the interrelationship between MPO, Mox-LDLs, and resolvins, highlighting a new perception of the role of Mox-LDLs in atherosclerosis.

## 1. Introduction

Atherosclerosis is a vascular disease characterized by chronic inflammation and the formation of atheromatous plaques in large and medium-sized arteries. Destabilization of plaques can lead to life-threatening events such as stroke and heart attack, so it is critical that the mechanisms involved in plaque development and rupture are well understood [1]. The clinical consequences of atherosclerosis are most likely determined by the inflammatory environment, which is the result of a balance between the pro-inflammatory and pro-resolving factors [2]. Vulnerable human atherosclerotic plaques have a lower pro-resolving/pro-inflammatory ratio than stable plaques, and the restoration of this balance promotes plaque stability in mice [3]. Therefore, the progression of atherosclerosis may be linked to a defective inflammation–resolution process. The lipid mediators involved in inflammation resolution are called specialized pro-resolving mediators (SPMs) [4]. The production of SPMs enables the body to compensate for the deleterious effects of inflammation. SPMs are rapidly produced at the onset of inflammation and co-exist with pro-inflammatory signals. The balance between these pro-inflammatory signals and SPMs determines the duration and intensity of inflammation [5].

The role of myeloperoxidase (MPO) in atherosclerosis has been investigated for several decades. MPO is considered a biomarker for various chronic inflammatory diseases, including cardiovascular disease and rheumatoid arthritis. In addition, plasma MPO levels may be related to increased risk, severity, and extent of coronary artery diseases (CAD) in patients [6,7]. Elevated plasma MPO concentrations have been reported in patients with CAD, unstable angina, and acute myocardial infarction [6,8]. The involvement of MPO in cardiovascular diseases is often related to its pro-oxidant activity. MPO is one of the main physiological ways by which low-density (LDL) and high-density (HDL) lipoproteins can be oxidized [9,10]. Oxidized LDLs and HDLs are involved in the development and progression of atheromatous plaques and inflammation [11]. MPO oxidized-LDLs (Mox-LDLs) have multiple effects on cells involved in the development of atherosclerosis. They induce the production of pro-inflammatory mediators, interleukin-8 (IL-8) and tumor-necrosis factor-α (TNF-α), by endothelial cells and macrophages respectively [12]. In this review, we discuss the involvement of MPO and Mox-LDLs in atherosclerosis, and, more specifically, their role in modulating inflammation and its resolution, with a special focus on the resolution of endothelial inflammation.

## 2. Myeloperoxidase

MPO is a heme-containing enzyme that was first known for its role in the body’s defense against pathogens. This enzyme is mainly found in azurophilic granules of neutrophils and is released when the neutrophils are activated. It can also be found in the lysosomes of monocytes [13]. MPO expression is not, however, restricted to neutrophils and monocytes. For example, endothelial cells can produce MPO when oxidative stress is induced by non-lethal doses of hydrogen peroxide (H_2_O_2_) [14]. MPO uses H_2_O_2_ and chloride ions (Cl^−^) to produce hypochlorous acid (HOCl) [15]. HOCl is a powerful oxidant and participates in the microbicidal activity of MPO by oxidizing molecules from the pathogen membrane. Some other (pseudo-)halogenous oxidants can be synthesized by MPO, such as hypobromous acid (HOBr) or hypothiocyanous acid (HOSCN) [16]. HOCl is thought to be the most abundant product because of the availability of chloride ions in the extracellular medium, but it should be noted that thiocyanate (SCN^−^) is a better substrate for MPO and that in patients with high SCN^−^ levels, such as smokers, the concentrations of HOSCN produced will be significantly higher [17]. As a matter of fact, MPO has several roles in major events in atherosclerosis such as endothelial dysfunction, atherosclerotic plaque destabilization, and lipoprotein oxidation (see below).

### 2.1. Endothelial Dysfunction

MPO seems to act at all stages of atherosclerosis. A correlation has been established between MPO activity and endothelial dysfunction. Cheng et al. [18] pharmacologically inhibited MPO in mice and observed improved endothelial function in two different inflammatory models. The endothelium reacts as a selective barrier, regulating vascular tone, angiogenesis, inflammation, and platelet activation and adhesion. When the endothelium no longer fulfills its functions correctly, endothelial dysfunction occurs [19]. Nitric oxide (**^•^**NO) is produced by endothelial NO synthase (eNOS) in endothelial cells and is a key mediator in the maintenance of homeostasis [20]. By reducing **^•^**NO bioavailability, MPO promotes endothelial dysfunction. First, **^•^**NO can be directly consumed by MPO [21]. Second, the production of **^•^**NO is reduced in the presence of MPO and its products. Indeed, in vitro, HOCl causes an uncoupling of eNOS that makes the production of **^•^**NO impossible [22]. HOCl also chlorinates L-arginine, the physiological substrate of eNOS for **^•^**NO production and the chlorinated L-arginine inhibits endothelial **^•^**NO synthesis in a rat model [23]. In vitro, MPO decreases the phosphorylation of eNOS via a pathway involving the activation of µ-calpain. The phosphorylation of eNOS is essential for its **^•^**NO production. This same MPO/calpain pathway is also involved in the increase of vascular cell adhesion molecule 1 (VCAM-1) expression by endothelial cells, which induces an increase in leukocyte adhesion to the endothelium [24].

### 2.2. Plaque Destabilization

MPO is also involved in plaque destabilization via the production of HOCl. During the development of the atheromatous plaque, smooth muscle cells migrate to the intima and synthesize extracellular matrix components to form a fibrous cap that isolates the plaque lipid core from the bloodstream [25]. HOCl could act on the fibrous cap by modulating matrix metalloproteinase (MMPs) activity. MMPs degrade the extracellular matrix components of the plaque such as collagen. In vitro, HOCl activates MMP-7, neutrophils collagenase (MMP-8), and gelatinase (MMP-9) and inactivates the tissue inhibitor of metalloproteinases (TIMP-1) [26,27,28,29]. It also causes apoptosis and the detachment of endothelial cells in vitro and could therefore erode the plaque in vivo [30]. This action of MPO on MMPs is consistent with the research of Rashid et al. [31], who found that MPO-deficient mice had more stable plaques with thicker fibrous caps than did the control group. These authors also showed that MPO activity was more intense in less stable plaques [31].

In the last decade, a new heme-containing peroxidase has been identified, peroxidasin. Peroxidasin, previously called vascular peroxidase-1 (VPO1), is fundamental to the development of tissue basement membranes [32]. It allows the formation of collagen IV sulfilimine cross-links, which solidify the structure of the collagen IV network. Peroxidasin produces HOBr from H_2_O_2_ and Br^−^ and uses it as a reactive intermediate in the formation of sulfilimine cross-links [33]. In 2018, Zhang et al. [34] demonstrated that endothelial cells exposed to oxidized LDLs undergo programmed necrosis that was promoted by peroxidasin through a β-catenin pathway. Peroxidasin also has a pro-angiogenic role that could affect the outcome of atherosclerotic lesions [35]. The role of peroxidasin in atherosclerosis is not fully understood and further studies are needed to determine if peroxidasin impacts the development of atherosclerosis.

### 2.3. Lipoprotein Oxidation

One of the most studied roles of MPO in atherosclerosis is its involvement in the modification of lipoproteins. It has been almost 30 years since the presence of MPO in human atherosclerotic plaques was demonstrated by Daugherty et al. [36] and its role in lipoprotein oxidation has been studied ever since. Numerous studies have highlighted the ability of MPO to oxidize LDL and have confirmed the presence of these modified LDLs in human atherosclerotic lesions [37,38,39,40]. Oxidized LDLs play an important role in the development of the atheromatous plaque [41]. As previously stated, MPO generates HOCl, which reacts with molecules such as the protein moiety of LDL, apolipoproteinB-100 (apoB-100) [42]. MPO can bind LDLs. This binding is most likely mediated by ionic interactions because MPO is a cationic enzyme [43]. The liaison between MPO and LDL influences the oxidation of LDLs and increases MPO activity, contributing to the generation of Mox-LDLs [44]. Several studies have demonstrated a difference between HOCl-oxidized LDLs (HOCl-LDLs) and Mox-LDLs [44,45]. With its ability to interact with both LDLs and endothelial cells, MPO can oxidize LDLs in the bloodstream at the surface of the endothelial cell [46]. Depending on the degree of oxidation and apoB-100 modification, oxidized LDLs are no longer recognized by the LDL receptor. If the exact receptor(s) recognizing Mox-LDLs are not identified yet, an experiment with HOCl-LDLs showed that as the oxidant/liporoprotein molar ratio increased, interactions with the LDL receptor decreased [47]. Highly-oxidized LDLs, are, however, recognizable by scavenger receptors such as CD36, SR-A, and LOX-1 [48,49,50]. These receptors are located on the surface of macrophages and permit the unrestricted internalization of oxidized LDLs [51]. In vitro, HOCl-LDLs are internalized by THP-1 macrophages through class B scavenger receptors SR-B1 and CD-36 [52]. The latter also recognizes LDLs that are oxidized by an MPO-H_2_O_2_-NO_2_^−^ system [53]. The accumulation of intracellular lipids eventually transforms the macrophages into foam cells. These foam cells accumulate in the intima and form the fatty streaks that are the first visible lesions of atherosclerosis. The fatty streaks then evolve into an atherosclerotic plaque [25]. It is noteworthy that MPO is also able to oxidize HDLs [10,54]. The cholesterol efflux process, which plays a protective role in atherosclerosis, is impaired when HDLs are oxidized by MPO. By oxidizing apolipoprotein A-I in HDLs, MPO causes HDLs to lose their protective functions in atherosclerosis and become pro-inflammatory [54]. 

Interestingly, Liu et al. [55] found that oxidized LDLs influence endothelial cell production of MPO. They also observed increased MPO expression and endothelial senescence in a hyperlipidemic rat model. The upregulation of MPO by oxidized LDLs may be related to the accelerated endothelial senescence through a mechanism involving the β-catenin/p53 pathway [55]. The problem with the term “oxidized LDL” is that it represents a multitude of different oxidized LDLs. Currently, the exact mechanisms involved in LDL oxidation in vivo are still unknown. There are many possibilities for oxidation, including oxidation mediated by ROS, by metal ions, or by enzymes such as MPO. Many experimental models use copper-oxidized LDLs, but elevated levels of metal ions like copper are only found in advanced plaques, whereas MPO is present early in plaque development [56,57]. Various effects have been demonstrated depending on which oxidized LDLs (Mox-LDLs versus copper-oxidized LDLs) are used. Therefore, it is necessary to be cautious about the application of oxidized LDL data to Mox-LDLs [58].

## 3. Mox-LDL-Cell Interactions

### 3.1. Mox-LDL-Macrophages

In addition to their role in the formation of foam cells after intracellular accumulation, Mox-LDLs interact with various cell types involved in the development of atherosclerosis, such as macrophages and endothelial cells. When THP-1 cells were incubated with Mox-LDLs, an increase in TNF-α secretion was observed [51]. Calay et al. [59] studied the consequences of increased oxidative stress in macrophages. They incubated RAW264.7 and peripheral blood mononuclear cell (PMBC)-derived macrophages with Mox-LDLs and confirmed an intracellular increase in ROS, which was mediated by cytosolic phospholipase A2 (cPLA2). This increase was associated with the activation of the NF-E2-related factor 2 (Nrf2) transcription factor, which triggers an antioxidant response by regulating the expression of genes involved in the protection against oxidative stress, such as glutamate-cysteine ligase (Gclm) and heme oxygenase-1 (HO-1) [59]. These data illustrate one of the mechanisms by which the body maintains a constant balance between oxidative stress and anti-inflammatory response. 

Mox-LDLs play a role in macrophage polarization. Macrophages express different phenotypes depending on their environment. These phenotypes are generally divided into two groups: M1 macrophages, which are pro-inflammatory, and M2 macrophages, which are anti-inflammatory. The M1-M2 classification is, however, inadequate because it represents two extremities, whereas macrophages probably have more nuanced phenotypes. When unpolarized RAW264.7 macrophages were incubated with Mox-LDLs, the expression of the marker genes increased for both M1 and M2 macrophages, but these Mox-LDL-stimulated macrophages tended to have an intermediate phenotype, which was more anti-inflammatory [60].

### 3.2. Mox-LDL-Endothelial Cells

Endothelial cells maintain a balance between coagulation and fibrinolysis. Incubation with Mox-LDLs disrupts the fibrin clot elimination at the endothelial cell surface. A hypofibrinolytic state has been demonstrated in patients with atherosclerosis and the accumulation of fibrin at the endothelial cell surface increases endothelial permeability, which enables the accumulation of lipids in the subendothelial space and the formation of foam cells [61]. Oxidized LDL has often been considered a source of oxidative stress, and, indeed, copper-oxidized LDLs induce oxidative stress in endothelial cells [62]. However, as previously stated, copper-oxidized LDLs and Mox-LDLs have different effects, and it has been shown that Mox-LDLs do not induce ROS production by human aortic endothelial cells (HAEC) [63]. Another effect of Mox-LDLs on the endothelium has been demonstrated in vitro on angiogenesis. The interaction of Mox-LDLs with endothelial cells decreases their motility, their tubulogenesis, and their migration [64].

While studying the effect of Mox-LDLs on endothelial cells, Dufour et al. [65] noticed that the extracellular concentration of resolvin D1 (RvD1) increased rapidly after stimulation with Mox-LDLs and native LDLs (Nat-LDLs) with a synergistic action in these LDLs. These effects were not observed with non-physiological copper-oxidized LDLs or MPO alone. RvD1 is one of the many SPMs that play a role in the resolution of inflammation. 

Little is known about the mechanisms involved in the release of RvD1 after stimulation by Mox-LDLs. Dufour et al. [65] demonstrated the role of phospholipase A2 (PLA2) and ROS in the production of RvD1. The relationship between Mox-LDLs and RvD1 needs to be further studied given that the only existing information at the moment comes from an in vitro model on endothelial cells [65]. The hypothesis that the stimulation of SPM production by Mox-LDLs may help compensate for the deleterious effect of Mox-LDLs in vivo remains to be proven. The pathways by which Mox-LDLs interact with endothelial cells are largely undetermined and appear to be diversified. Recently, El-Hajjar et al. [66] demonstrated that Mox-LDLs exert pro-inflammatory activity on HAECs by stimulating IL-8 production through the scavenger receptor LOX-1. In contrast, RvD1 production by endothelial cells is not dependent on LOX-1 and involves other pathways that are not fully understood [67].

There are only a few articles that tend to demonstrate a possible protective effect of Mox-LDLs and MPO on the clinical manifestations of atherosclerosis, and none of the studies have been conducted in humans. In 2001, Brennan et al. [68] studied atheromatous plaques in MPO-deficient mice and found that these mice had larger plaques than the control group. The authors suggested a possible protective role for MPO in atherosclerosis, but the murine model has its limitations and there are some differences between murine and human atheromatous lesions—i.e., MPO levels are five- to tenfold higher in humans than in mice and MPO may not be catalytically active in the mouse artery wall [68]. Recently, a study demonstrated a decrease in neutrophil adhesion and migration in an inflammation model, resulting from the binding of MPO to neutrophils. These data were validated in vitro and in vivo in mice and the authors have concluded that MPO acts as an inhibitor of neutrophil migration during inflammation. This proposes a possible protective role for MPO in inflammation [69]. These data must be balanced by another study by Klinke et al. [70], which shows that neutrophil recruitment is MPO-dependent in mice. This effect of MPO on neutrophil attraction would be related to its cationic charge and not to its enzymatic activity. A summary of the interactions of Mox-LDLs with macrophages and endothelial cells can be found in Figure 1. Further research is needed to understand the underlying mechanisms involved in the dual role of Mox-LDLs and MPO in human atherosclerosis plaques and inflammation, but the production or liberation of RvD1 by Mox-LDLs-stimulated endothelial cells may be a reasonable starting point. However, more in vivo and human data are needed before any conclusions can be drawn.

## 4. Bioactive Lipid Mediators: Implications in Atherosclerosis

Pro-inflammatory mediators, such as prostaglandins or leukotrienes, are produced by immune cells in the first stages of acute inflammation. Those mediators are synthesized very rapidly from arachidonic acid by cyclooxygenases or lipoxygenases, within seconds of the onset of inflammation. Depending on the cytokines and signals perceived in the environment, leukocytes can move from the production of these pro-inflammatory mediators to the production of pro-resolving mediators, such as lipoxins and resolvins, resulting in a temporal switch in mediator profiles [71]. This process generates lipid mediators by successive oxidations from n-6 polyunsaturated fatty acid (PUFA) and from n-3 PUFAs [72]. Lipoxin, resolvin, protectin, and maresin are part of the so-called SPMs [73].

Extensively studied since 2000, these SPMs are potent inhibitors of human polymorphonuclear (PMN) leukocyte transendothelial migration and infiltration at picogram-to-nanogram concentrations. They enhance monocyte/macrophage clearance of cellular debris and apoptotic PMNs, which is required for tissue homeostasis [74,75]. Lipoxin A4, a SPM derived from arachidonic acid, downregulates azurophilic degranulation of stimulated PMNs and, by doing so, decreases MPO activity [76]. As RvD1 is produced in response to stimulation by Mox-LDLs, we will detail its production and actions in atherosclerosis hereafter.

RvD1 is a bioactive molecule derived from docosahexaenoic acid (DHA). After oxidation by 15-lipoxygenase (15-LOX) at the 17th chain’s carbon to form 17S-H(p)DHA, this intermediate is further oxidized by 5-lipoxygenase (5-LOX) on the 7th carbon to give 7S-hydroperoxy-17S-HDHA. This latter intermediate is converted into epoxide (7S, 8-epoxy-17S-HDHA) and then undergoes enzymatic hydrolysis to obtain RvD1 [77]. RvD1 acts locally at the site of inflammation by binding to two receptors: the lipoxin A4 receptor, ALX, and an orphan receptor coupled to a G-protein, the GPR32. Both receptors are found on the surface of phagocytes. When RvD1 binds to these receptors, it promotes phagocytosis, which is essential for the resolution of inflammation [78]. RvD1 also stimulates the synthesis of another SPM, lipoxin B4, and decreases that of leukotriene B4 (LTB_4_) in an ALX-dependent manner in mice [79]. 

Various aspects of RvD1 being potentially involved in the modulation of atherosclerosis have been studied. First, the level of RvD1 in highly necrotic atherosclerotic plaques was shown to be defective in humans and these plaques showed a marked imbalance between RvD1 and LTB_4_ [3]. The RvD1:LTB_4_ ratio measured in human salivary samples was proposed as a potential predictor of atherosclerosis, taking into account the correlation observed with carotid intima media thickness [80]. Second, interventional studies were conducted to evaluate the effects of RvD1. When neutrophils were exposed to RvD1, interactions with the endothelium decreased significantly in a concentration-dependent manner [78]. In addition, RvD1 decreased the expression of adhesion molecules in inflammatory conditions, such as vascular cell adhesion molecule 1 (VCAM-1), which decreases the interactions of macrophages with the endothelium [81]. Treatment with RvD1 also increased pro-resolving markers, such as arginase 1 and mannose receptor C-type 1, in macrophages and regulated leukocyte function and plasticity [82]. The effects of RvD1 on endothelial permeability have also been explored. RvD1 maintained endothelial cell integrity by preventing lipopolysaccharide-mediated barrier functions [83]. RvD1 enhanced efferocytosis, a critical cellular process of inflammation resolution that is defective in atherosclerosis [84,85]. Figure 1 summarizes the effects of RvD1 on neutrophils, macrophages, and endothelial cells.

To determine which cells produce RvD1, Chatterjee et al. [86] performed different experiments on isolated human arteries and on cell cultures. During the experiments, endothelial cells or smooth muscle cells produced RvD1 when exposed to its precursor, 17-HDHA. Arterial segments, with and without DHA or 17-HDHA supplementation, also generated RvD1. As seen earlier, 5-LOX is essential for the production of RvD1 at the endothelial and smooth muscle cell level. It was noted that both types of cells constantly expressed 5-LOX and that its expression could be strongly stimulated in endothelial cells in the presence of 17-HDHA [86]. Depending on its location in the cell, 5-LOX promotes the synthesis of RvD1 or pro-inflammatory leukotrienes. Cytoplasmic 5-LOX and nuclear 5-LOX are responsible for the synthesis of SPMs and leukotriene, respectively [87]. RvD1 induced a functional switch toward a pro-resolving phenotype in isolated primary human macrophages [88]. As in endothelial cells, RvD1 triggered a positive feedback loop by the translocation of nuclear 5-LOX to the cytoplasmic compartment in human macrophages with secondary reduced production of pro-inflammatory LTB_4_ and increased levels of lipoxin A4 (LXA4), another SPM [87]. Various actions of RvD1 have led to the development of novel interventional strategies such as a biodegradable polymer wrap loaded with RvD1 that allows for local and long-lasting delivery of RvD1. The local perivascular delivery of RvD1 attenuated neointimal hyperplasia in a rat model of carotid angioplasty [89]. Table 1 summarizes the effects of RvD1 discussed herein.

In summary, there are numerous pathways of RvD1 production to assure its generation. Depending on the cell they interact with, Mox-LDLs induce endothelial RvD1 biosynthesis [65]. Although Mox-LDLs are known to be pro-inflammatory, they are therefore also pro-resolving.

Few articles have examined the relationship between SPMs and MPO [65]. Some interventional studies using n-3 PUFA supplements have been conducted. Poreba et al. [90] found no effects of n-3 PUFAs on MPO or RvD1 level in patients with type 2 diabetes and established atherosclerosis. However, Barden et al. [91] observed that supplementation with n-3 PUFA led to a significant reduction in plasma MPO in patients with chronic kidney disease. The effects of bioactive lipids on MPO reduction should be explored.

**Table 1 antioxidants-11-00874-t001:** Effects of SPMs on various cell types.

SPM	Cell Type	Effect	Reference
RvD1	Macrophage	Stimulation of phagocytosis	Krishnamoorthy et al. [78]
	Macrophage	Enhanced efferocytosis	Rymut et al. [84]
	Macrophage	Polarization of primary macrophages and repolarization of previously polarized M1-macrophages to a pro-resolution phenotype	Schmid et al. [88]
	Macrophage	Switch from a M1 phenotype to a pro-resolution M2-like phenotype	Dalli et al. [92]
	Neutrophil	Decreased actin polymerization which is essential to neutrophil migration. Limited infiltration and transendothelial migration	Krishnamoorthy et al. [78]
	Cardiac fibroblast	Decreased expression of adhesion molecules ICAM-1 and VCAM-1	Salas-Hernández et al. [81]
	Macrophage	Decreased production of pro-inflammatory cytokines. Increased expression of pro-resolving markers. Polarization of macrophages toward a pro-resolving phenotype	Kain et al. [82]
	Endothelial cell	Protection of endothelial cell adherens junction. Maintenance of endothelial barrier integrity and permeability	Chattopadhyay et al. [83]
	Macrophage	Translocation of 5-LOX from the nucleus to the cytoplasm, inducing increased synthesis of LXA4 and decreased synthesis of LTB_4_	Fredman et al. [87]
17-HDHA/RvD1	Endothelial cell and VSMC	Translocation of 5-LOX from the nucleus to the cytoplasm	Chatterjee et al. [86]

Abbreviations: RvD1: resolvin D1, ICAM-1: intercellular adhesion molecule 1, VCAM-1: vascular cell adhesion molecule 1, 5-LOX: 5-lipoxygenase, LXA4: lipoxin A4, LTB_4_: leukotriene B4, 17-HDHA: 17-hydroxy-docosahexaenoic acid, VSMC: vascular smooth muscle cell.

## 5. Monocyte-Derived Macrophages and SPMs: Interplay in Atherosclerosis

Macrophages play a critical role in atherosclerosis progression, responding to various environmental signals, including bioactive lipid-derived, and adapting their phenotype over time. Macrophage phenotype diversity and its role in atherosclerosis were reviewed recently by Jinnouchi et al [93]. In human atherosclerotic lesions, macrophage subpopulations have been identified and were more numerous in symptomatic plaques [94,95]. Cho et al. [95] showed that M1 macrophages were dominant within unstable carotid plaques.

Lipoproteins accumulate in the intima and are ingested by macrophages. After the engulfment of lipoproteins, especially modified lipoproteins, the activation of toll-like receptors (TLR), INF-γ or IL-1β, polarize macrophages into the M1 phenoytpe [96]. Thus, M1 macrophages are involved in producing pro-inflammatory cytokines such as IL-6, IL-12, and TNF-α, whereas M2 macrophages have an opposite role and produce anti-inflammatory cytokines like IL-10 and transforming growth factor- β (TGF-β) [88]. Pireaux et al. [97] highlighted the importance of the disease-related environment in macrophage polarization. They observed a higher percentage of M2 monocytes in the plasma of hemodialysis patients than in the controls. In contrast, the concentrations of inflammatory biomarkers—such as CRP, macrophage-colony stimulating factor (M-CSF), and IL-8, as well as chloro-tyrosine and homocitrulline—which are markers of myeloperoxidase-associated oxidative stress, were positively correlated with the M2 phenotype [97]. RvD1 was negatively correlated with M1 macrophage levels but not with M2 levels in patients undergoing hemodialysis [65]. These results suggest a more complex interaction of pro-inflammatory and pro-resolving mediators in self-regulation and macrophage polarization than expected. Indeed, RvD1 and the intermediary 17-HDHA promote polarization toward an anti-inflammatory M2-phenotype [92,98]. Furthermore, other lipid mediators, such as nitrosylated-fatty acids and n-3 PUFA products, polarize plaque macrophages toward anti-inflammatory and pro-resolving phenotypes, thereby confirming the complex dual role of these cells [99].

In the environment, macrophage phenotype development is under specific transcriptional control. For example, the NFκB pathway favors M1 production whereas peroxisome proliferator–activated receptor γ (PPARγ) favors M2 [100,101]. A distinct phenotype, the Mox-macrophage, was identified when oxidized phospholipids activated stress response genes via Nrf2 [102]. The denomination “Mox macrophage” and “Mox-LDL” can be confusing in the sense that “Mox macrophages” do not appear to be related to MPO or Mox-LDLs. Macrophages possess distinct lipid mediator profiles according to their phenotype. Indeed, the comparison of endogenous mediator biosynthesis in M1 and M2 macrophages provides some insight. M2 macrophages produce maresin 1 and RvD5, both derived from DHA, as well as resolvin E2 (RvE2) and lipoxins. Conversely, M1 macrophages biosynthesize cyclooxygenase-derived lipid mediators such as prostaglandin E2, prostaglandin F_2α_, and thromboxane B2 [103]. Whether the lipid mediator profile defines the distinct functions and phenotypes of macrophages or is the result of the activity of macrophages deserves further investigation. In an experiment conducted by Dalli et al. [103], the phagocytosis of apoptotic PMNs by macrophages—in other words, efferocytosis—stimulated SPM production. Of interest, the activation of the MerTK receptor, a macrophage receptor involved in the process of efferocytosis, upregulated SPM production by increasing the cytoplasmic/nuclear ratio of a key SPM biosynthetic enzyme, 5-LOX [104]. This finding indicates that lipid mediator enzymes have a significant role in this process, and they will be briefly described below.

Phospholipases and lipoxygenases work in concert to produce both pro-inflammatory and pro-resolving lipid mediators. Cytosolic and calcium-independent phospholipase A2 isoforms are ubiquitously expressed and activated in acute inflammation, generating lipid mediators from membrane phospholipids [105]. Arachidonic acid, for example, is transferred via 5-lipoxygenase activating protein (FLAP), a protein activating 5-LOX, which initiates the synthesis of pro-inflammatory LTB_4_ and leukotriene C4 [106]. A specific PLA2 linked to lipoprotein (lp-PLA2) is an enzyme formed by macrophages and foam cells in atherosclerotic plaques [107]. This enzyme is responsible for the hydrolysis of oxidized LDL particles and the subsequent release of pro-inflammatory lipids.

More specifically, the induction of cPLA2 was investigated by Calay et al. [59] in THP-1, a cellular model of macrophages. These authors found that high concentrations of Mox-LDL increased the activity of cPLA2, thereby releasing fatty acids that are stored in the cell membrane. Of interest, secondary ROS production by cPLA2 was induced only by Mox-LDL and not by ox-LDL, for which the production of ROS depends on NADPH oxidase [59]. Moreover, adding a PLA2 inhibitor (MAFP) reduced RvD1 production by human microvascular endothelial cells [65]. This leads us to presume the duality of this enzyme isoform, which is involved with both inflammatory *and* pro-resolving mediators, by making the necessary substrate available. In addition, by generating pro-inflammatory lipid mediator and ROS signals, calcium-independent PLA2 (iPLA2) affects macrophage polarization. The reduced activity of iPLA2 attenuates M1 polarization and promotes M2 polarization [108].

## 6. Conclusions

Atherosclerotic plaques are a battlefield of pro-inflammatory and pro-resolving factors with a particular interplay of monocyte-derived macrophages. MPO is one of the leading agents inducing oxidative stress and is the basis for Mox-LDL generation. These particular LDLs, which are closer to physiological conditions than ox-LDLs, have a double-sided effect in inflammation. Mox-LDLs modulate the SPM/pro-inflammatory balance by promoting RvD1 and IL-8 at the endothelial level, suggesting a crucial role in the development and stabilization of the atheromatous plaque. During oxidative stress regulation, the induction of SPMs and the M2 macrophage phenotype is a matter of survival. The defective resolution of inflammation at all stages of vascular inflammation has led to the proposal of the resolvin/leukotriene ratio as a relevant biomarker in atherosclerosis. In addition, SPMs such as resolvin D1 successfully prevented atheroprogression in pre-clinical models. However, the role of MPO, and, more specifically, Mox-LDLs in the production of SPMs by endothelial cells and macrophages is not well established and opens the field to future research on their roles in the resolution of inflammation.

## Figures and Tables

**Figure 1 antioxidants-11-00874-f001:**
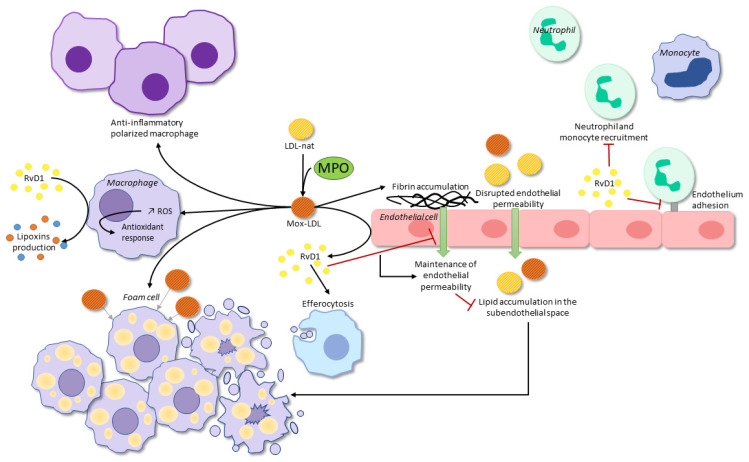
Effects of Mox-LDLs on macrophages and endothelial cells. At the center of the figure: MPO oxidation products transform native LDLs (Nat-LDLs) to MPO-oxidized LDLs (Mox-LDLs). Left: Mox-LDLs can either be engulfed by macrophages and participate in the formation of foam cells or interact with macrophages. They influence the polarization of macrophages toward an anti-inflammatory phenotype and induce an increase in intracellular reactive oxygen species (ROS), which leads to a compensatory response by stimulating the antioxidant response of macrophages. RvD1 stimulates the production of lipoxins, another SPM, by macrophages. Right: Mox-LDLs also disrupt fibrin elimination at the surface of endothelial cells, which results in an accumulation of lipids, such as Nat-LDLs or Mox-LDLs, into the subendothelial space. When they interact with endothelial cells, Mox-LDLs stimulate the release of resolvin D1 (RvD1), a specialized pro-resolving mediator (SPM). RvD1 stimulates efferocytosis, which is essential to the resolution of inflammation and return to homeostasis, maintains endothelial cell permeability, and decreases the recruitment and adhesion of leukocytes to the endothelium.

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
