# Peer review of "Unexpected Role of MPO-Oxidized LDLs in Atherosclerosis: In between Inflammation and Its Resolution"

_antioxidants, 2022, doi:10.3390/antiox11050874_

Round 1
Reviewer 1 Report
This manuscript reviews some of the literature on the role of myeloperoxidase (MPO)-modified/oxidized low density lipoproteins in the development and progression of atherosclerosis.
The manuscript essentially is split into two parts – the first on MPO induced modifications and the second part which is almost completely a review of resolvins and related molecules. These two parts are not very well integrated. The first section covers work that has been reviewed or discussed in the literature previously including in other papers by the same group (e.g. Vanhamme, L., Zouaoui Boudjeltia, K., Van Antwerpen, P., & Delporte, C. (2018). The other myeloperoxidase: Emerging functions. Archives of biochemistry and biophysics, 649, 1-14. doi:10.1016/j.abb.2018.03.037), and there is limited new information in this section.
This section is rather selective and poorly referenced with some of the seminal work in this field not (or poorly) referenced (e.g the work of Stan Hazen’s group, Alan Daugherty, Stephan Baldus/Anna Klinke, Ernst Malle, Hazell and Stocker etc).
There are a number of statements that are open to dispute or where references to support the statements are not provided. The coverage is rather unbalanced and incomplete, and there are some areas in which the authors are over-extrapolating from cell studies to human or animal disease, which is not warranted by the available data.
Examples include:
Line 24: ‘Mox-LDLs have a protective effect in atherosclerosis’ – where is the animal or human evidence for this ?
Line 38/39: ‘Progression of atherosclerosis is linked to a defective inflammation-resolution process’. What is the in vivo evidence for this ?
Line 85/86: reference needed – Eiserich/Baldus
Line 99: references to action of HOCl activating MMPs and deactivating TIMPs. In vivo evidence for this ?
Lines 107-121: This section needs updating. VPO1 is now called peroxidasin and there are multiple other studies (e.g. from the Billy Hudson group) on this point. It is well established that this enzyme does not generate significant levels of HOCl, with the original data due to contamination of the NaCl samples used with bromide ions. This enzyme predominately generates HOBr, and its role is in crosslinking of the NC1 domains of collagen in the ECM.
Lines 126-7 and also 134-136: ref 26 is not the most appropriate reference here. There are many earlier seminal papers on this topic which should replace ref 26.
Line 130: why should the interaction of MPO with LDL make Mox-LDLs unique ?
Line 141-2: there are many other important refs missing here.
Line 154: Refs for presence of metal ions in advanced lesions – e.g. N. Stadler
Section 3: some of the work of Lars Leichart and others working on the effects of HOCl/MPO modified proteins could/should be discussed here
Line 210: The data reported in ref 41 do not address the in vivo situation – they are model studies on endothelial cells. In the view of this reviewer there is no compelling evidence that Mox-LDLs do not cause cardiovascular events in vivo – particularly as it is extremely difficult – if not impossible – to ensure that there are only Mox-LDL present. The authors are overselling this data.
Line 277: subject
Line 281: 7th carbon to give…
Line 308/9: refs are needed here
Author Response
Thank you for your comments, they were very helpful. Please see attachment for the responses.

Reviewer 2 Report
In this review article, the authors discuss the pathophysiological roles of MPO, MPO oxidized LDL, and the release of resolvins. The review is well written, but unfortunately too unbalanced.
- The major weakness of this review article is that it is very unfocused. The concept of writing about potential anti-inflammatory aspects of MPO and Mox-LDL is valid. However, the article is based on only one original paper by the group (reference 41) showing that Mox-LDL induces resolvin production by endothelial cells. This is certainly interesting, but is discussed far too extensively; 51 of the 92 references revolve around this topic alone. This reviewer suggests that the interaction of MPO/ MPO oxidized LDL with endothelial cells should be discussed in much greater detail. In this context, it should also be fairly discussed that almost all studies showed pro-inflammatory effects of MPO and MPO oxidized LDL.
- On page 2, line 87 the authors write “Indeed, MPO, HOCl, and MPO-modified lipoprotein inhibit eNOS activity and HOCl chlorinates arginine, the physiological substrate of eNOS for NO production”. Here the authors only cite a review article, the original papers should be cited instead. This is just one example (of many), the authors should mainly cite original papers.The authors state on page 3, line 137 “The receptor implicated in the internalization of Mox-LDLs has not yet been determined [26]”. This is not correct. LDL oxidized by the myeloperoxidase (MPO)-H2O2-NO2 system is recognized by CD36 (doi.org/10.1074/jbc.M203318200) and LDL oxidized by the MPO product HOCl is recognized by CD36 and SR-BI (doi:10.1074/jbc.M308428200).
Author Response
Thank you for your comments which were very helpful and interesting. Please see the attachment for the responses.

Round 2
Reviewer 1 Report
Line 24: induces
Line 25: delete ‘in atherosclerosis’ as ‘in vitro’ is used at the beginning of this sentence
Lines 70 and also 71: H2O2 (subscript 2) in both cases
Lines 77 and 78: superscript negative sign for SCN-
Line 79: fact (singular)
Line 90 and elsewhere: superscript radical dot for NO.
Line 136: contributing to the generation…
Line 227: exert pro-inflammatory
Line 231: ‘that propose a possible protective’ may be better here
Line 387: the beginning of this line does not appear to make sense (‘from interventional studies…)
Author Response
Thank you for the corrections. Here is the point-by-point response to your comments :
Line 24: induces : corrected
Line 25: delete ‘in atherosclerosis’ as ‘in vitro’ is used at the beginning of this sentence : deleted
Lines 70 and also 71: H2O2 (subscript 2) in both cases: modified
Lines 77 and 78: superscript negative sign for SCN-: modified
Line 79: fact (singular): modified
Line 90 and elsewhere: superscript radical dot for NO.: modified
Line 136: contributing to the generation…: modified
Line 227: exert pro-inflammatory: modified
Line 231: ‘that propose a possible protective’ may be better here: modified: "These data were validated in vitro and in vivo in mice and the authors have concluded that MPO acts as an inhibitor of neutrophils migration during inflammation. This proposes a possible protective role for MPO in inflammation"
Line 387: the beginning of this line does not appear to make sense (‘from interventional studies…): modified : "Some interventional studies using n-3 PUFA supplements have been conducted."
Reviewer 2 Report
I have no further comments.
Author Response
Thank you for your feedback.